# Swelling characteristics and biocompatibility of ionic liquid based hydrogels for biomedical applications

Johanna Claus[1,2], Andreas Brietzke[3], Celina Lehnert[1,2], Stefan Oschatz[3], Niels Grabow[2,3], Udo Kragl[1,2]*

1 Department of Chemistry, Industrial and Applied Chemistry, University of Rostock, Rostock, Germany, 2 Department Life, Light and Matter, University of Rostock, Rostock, Germany, 3 Institute for Biomedical Engineering, Rostock University Medical Center, Rostock, Germany

* udo.kragl@uni-rostock.de

**Data Availability Statement:** All relevant data are within the manuscript and its Supporting Information files.

## Abstract

Polymers are commonly used in medical device manufacturing, e.g. for drug delivery systems, bone substitutes and stent coatings. Especially hydrogels exhibit very promising properties in this field. Hence, the development of new hydrogel systems for customized application is of great interest, especially regarding the swelling behavior and mechanical properties as well as the biocompatibility. The aim of this work was the preparation and investigation of various polyelectrolyte and poly-ionic liquid based hydrogels accessible by radical polymerization. The obtained polymers were covalently crosslinked with *N,N'*-methylenebisacrylamide (MBAA) or different lengths of poly(ethyleneglycol)diacrylate (PEGDA). The effect of different crosslinker-to-monomer ratios has been examined. In addition to the compression curves and the maximum degree of swelling, the biocompatibility with L929 mouse fibroblasts of these materials was determined in direct cell seeding experiments and the outcome for the different hydrogels was compared.

## Introduction

Since the 1960s, hydrogels are widely used in the medical and pharmaceutical industry as implants,[1] drug delivery-systems,[2,3] matrices for enzyme immobilization,[4] contact lens material,[5] tissue engineering and stent coatings.[6] Exploiting chemical and/or physical crosslinking enhances the mechanical properties such as stiffness, surface hardness, resilience to temperature and solvent attacks as well as swelling behavior.[7] Furthermore, hydrogels exhibit a variety of highly interesting properties such as outstanding biocompatibility, non-toxicity, biodegradability and a possible self-healing nature.[8,9] Those characteristics are frequently and broadly demanded and make such hydrogel based materials attractive for medical applications. Hence, a thorough control of the mechanical and swelling properties is crucial for tailor made products.[10]

A noticeable example was reported by Verma *et. al* describing the use of self-expanding, hydrophilic osmotic hydrogels for eyeball reconstruction in anophthalmia, to prevent the deformation of the skull.[11] The dry gels absorb the surrounding tissue fluid, swell up to ten times of their initial volume and build up the necessary counterpressure within the skull.[12]

**Funding:** We acknowledge financial support by Deutsche Forschungsgemeinschaft and University of Rostock within the funding programme Open Access Publishing. Funding by the Federal Ministry of Education and Research within RESPONSE "Partnership for Innovation in Implant Technology" (FKZ 03ZZ0910B) is gratefully acknowledged.

**Competing interests:** The authors have declared that no competing interests exist.

Compared to conventional, often rigid eye prostheses, frequently made of special glass, the shape flexibility of hydrogels, which grow simultaneously with the patient, offers an enormous advantage.[13] Moreover, a novel therapy of the degeneration of the intervertebral disc is of great interest in the medical community. Inserting a hydrogel stick into the interior of the intervertebral disc might be a much gentler method for the patient in comparison to a complete replacement of the disc, with a prosthesis consisting of different alloys, polyethylene, polyurethane and/or ceramics.[14] After complete swelling, the gel can relieve the weakened disc and support it in its functionality. Hence, being a minimal invasive procedure, the surgery is improved and can be performed in twilight sleep.[15]

However, recent research is still focusing on adjusting durability, degradability as well as mechanical properties and swelling behavior for more and more special fields of application. A current approach for hydrogel synthesis is the use of monomers derived from ionic liquids, which are polymerized to yield polyelectrolytes. These compounds are of special interest since the inherent properties can be easily fine-tuned by carefully selecting and combining different ionic functionalization to specifically exploit the advantages of both, ionic liquids and solid polymer structures, enhancing mechanical stability, improved processability and durability.[6] In addition to this, ionic liquids have an excellent ionic conductivity up to their decomposition temperature. The resulting hydrogels based on ionic liquids can be applicable as a conductor material.[16] Ohno *et al.*, for instance, described thermoresponsive polyelectrolyte and poly(ionic liquid) hydrogels with reversible water-uptake and release and aqueous salt solutions.[17,18] Annabi *et al.* investigated hydrogels based on poly(ethylene glycol) with adjustable degree of swelling and mechanical properties, showing potential for self-inflating tissue expanders.[19] Additionally, Bandomir *et al.* described the mechanical and thermal characteristics of imidazolium-based polymerized ionic liquids mainly focusing on the mechanical characterization.[10]

These examples illustrate only a fraction of the manifold adjustable properties of hydrogels and therefore possible applications and potential impact in the medical field. For the development of further innovations, as well as for the comprehension of transport and release mechanisms, a deeper understanding of the mechanical strength, physicochemical properties and biocompatibility is indispensable. Especially biocompatibility is a decisive obstacle in the approval process for medical and pharmaceutical applications and therefore serves as an early exclusion criterion for novel chemical compounds within the development process. In fact, there are only a few studies on natural compound-based hydrogels that focus not exclusively on chemical and physical but also on biological evaluation.[20] Furthermore, those studies were limited to cell viability studies on hydrogel eluates, which, however, represent only a part of the in vitro test methods crucial within the early approval process. In complement to eluate tests, approaches in which cell models were exposed in direct contact to the materials are also mandatory. Thus, another prior aim of this work was to develop and to establish a direct contact test procedure, including preparation of hydrogel specimen as well as quantitative analysis of the metabolic activity and microscopic imaging using L929 mouse fibroblasts as a cell model.

In summary, the tested hydrogels exhibit excellent biocompatibility in direct contact tests paired with outstanding mechanical performance. This is of significant relevance regarding biomedical applications. Our work did not only prove that no toxic residuals may leach out and disturb cell growth, but moreover that the evaluated hydrogel materials represent a suitable matrix for cell growth. Thus, polyionic liquid based hydrogel coatings support the formation of a full-covering cell layer, and may therefore have a mitigating influence on the foreign body reaction to implants. In combination with the mechanical properties and the ability to swell in water contact, hydrogel coatings are of special interest for implant applications such as stent or pacemaker coatings.

## Experimental section

### Materials and methods

**Chemicals.**    N,N'-methylenebis(acrylamide) (MBAA, 99%; Sigma-Aldrich), poly(ethylene-glycol)diacrylate (PEGDA, Mn = 250, 575, 700; Sigma-Aldrich); ethyleneglycoldimethacrylate (≥97%; TCI), *N,N,N* ,*N* -tetramethylethylendiamine (TMEDA, ≥99.5%; Sigma-Aldrich), ammoniumpersulfate (APS, 98%; Roth), 3-sulfopropylmethacrylate potassium (98%; Sigma-Aldrich), 3-sulfopropylacrylate potassium (Sigma-Aldrich), 2-acrylamido-2-methyl-1-propane-sulfonicacid (99%; Sigma-Aldrich), (vinylbenzyl)trimethylammonium chloride (99%; ACROS Organics), [2-(acryloyloxy)ethyl]trimethylammonium chloride (80 wt.% in $H_2O$; Aldrich), [2-(methacryloyloxy)ethyl]trimethylammonium chloride (75 wt.% in $H_2O$; Aldrich), 2-hydro-xyethylmethacrylate (97%; Alfa Aesar), [2-(methacryloyloxy)ethyl]dimethyl-(3-sulfopropyl) ammoniumhydroxide (95%; Sigma-Aldrich), 2-methacryloyloxyethylphosphorylcholine (97%; Sigma-Aldrich), 1-vinylimidazole (≥99%; Alfa Aesar), bromoethane(≥99%; Sigma-Aldrich), amino-2-propanol(93%; TCI), methacrylicanhydride (94%; Alfa Aesar) were all used as received without further treatment. 1-Vinyl-3-ethyl-imidazoliumbromide (VEtImBr) and 2-Hydroxypro-pyl methacrylamide (HPMAA) were prepared according to published procedures.[21,22]

**General procedure for the syntheses of hydrogels containing polyelectrolyte mono-mers.**    All polyelectrolyte hydrogels were synthesized by radical polymerization. The monomer and the related amount of the crosslinker (2 mol% in respect to the monomer) were dissolved in deionized water, adjusting the total monomer concentration to 2 mol/L. Subsequently, APS solution was added (0.1 mol% in degassed water in respect to monomer amount). The reaction solution was degassed for 15 min and the required amount of the TMEDA solution was added (1.9 mol% of total monomer amount in degassed water). After a short mixing, the reaction mix-ture was poured into cylindrical molds (10 mm height × 6 mm diameter), closed airtight and let rest for 24 h at 22 ± 2˚C. Elemental analyses and ATR measurements were carried out for all hydrogels and are listed in the Supporting Information.

**Mechanical characterization.**    The hydrogels were deformed by compression at 22 ± 2˚C with a single-column zwicki-line testing machine Z 2.5 [Zwick/Roell] equipped with a 50 N load cell and jaw inserts. All measurements were performed in triplicates and analyzed with *testXpert* standard. The synthesized cylindrical hydrogels were subsequently compressed with a speed of 2 mm/min.

**Evaluation of swelling behavior.**    Solvent uptake kinetics of freshly synthesized hydrogels were studied gravimetrically in water at 37˚C as a function of time in a strainer. The weights of the swollen gels were determined at different intervals, after dripping through the strainer and dabbing the samples off, until the equilibrium swelling was attained. The degree of swelling was calculated according to the following equation:

$$q_t = \frac{W_t}{W_0} - 1$$

$W_0$ is the initial weight and $W_t$ the final weight of the gel after the time t. Each degree of swelling is reported as an average of three separate measurements and within standard deviation.[23]

### Biocompatibility

**Preparation of hydrogel specimen.**    For biocompatibility testing, 1 mL of the monomer-crosslinker solution was polymerized between two 50 mm diameter cover slips to obtain specimen with the maximum possible surface. Furthermore, the capillary force between cover slips ensured a minimal layer thickness, which is a basic requirement for cell culture

and microscopy. Hydrogels were swollen in deionized water (10 mL) and washed with a minimum of three water exchanges within 72 h. Due to the correlation of swelling behavior and ion concentration, the hydrogels were put into a phenol red free cell culture medium over night to avoid volume changes during the cell cultivation. For screening test n = 4 specimen of 6 mm diameter were punched out and put on a 96-well microtiter plate.

**Cell culture and cell viability assay.** L929 mouse fibroblasts (CCL-1, ATCC) were cultured in Dulbecco's Modified Eagle Medium (DMEM, PAN BIOTECH, Aidenbach, Germany) with 4.5 mg Glucose, 10% fetal calf serum (FCS), 1% Penicillin/Streptomycin and 3.7 g/L $NaHCO_3$. $1x10^4$ L929 mouse fibroblasts were seeded directly on the hydrogel specimen with 200 μL culture medium per well and incubated under cell culture conditions (37˚C, 5% $CO_2$) for 46 hours (fibroblast cell density 62500 cellsxcm$^2$). To assess relative cell viability, a resazurin based CellQuanti-Blue Cell Viability Assay Kit (BioAssay systems, Hayward, CA, USA) was implemented. 10% CellQuanti-Blue supplement was added to the wells followed by an incubation of another 2 hours under same conditions. The metabolic turnover from resazurin to the fluorescent resorufin (absorption 544 nm, emission 590 nm) was detected with the Fluostar optima (BMG LABTECH, Ortenberg, Germany). Replicates (n $\geq$ 3) were tested for normal distribution and subjected to a Nalimovs' test for outliers. Means and standard errors (SEM) were calculated in Sigma Plot.

**Cytochemical analysis and microscopic imaging.** For the evaluation of cell morphology and cell adhesion light microscopy was carried out with Nikon Eclipse TS100 inverted microscope and Nikon digital sight DS-Fi1 digital camera. Cells were stained with Calcein AM (Thermo Fisher Waltham, MA, USA) to prove the viability of the living cells. Calcein AM is activated via hydrolyses by intracellular esterase and binding to calcium, which in turn leads to the emission of fluorescence (515 nm). The general procedure for the treatment of the cells includes one washing step with PBS and 30 min staining with a 1:200 Calcein AM staining solution in PBS. Imaging was realized with an Olympus IX81 confocal laser scanning microscope (CLSM, for Calcein staining 488 nm excitation and 505–605 nm fluorescence imaging filters).

## Results and discussion

### Synthesis

All hydrogels used in this study were synthesized by free radical polymerization starting from different monomers (Fig 1), using MBAA and PEGDA with various chain lengths as crosslinker and APS/TMEDA as initiator system.

A general reaction scheme can be found in Fig 2. APS and TMEDA form two main types of primary initiating radicals through an initiation mechanism via a charge transfer complex and a cyclic transient state. The initiation of the vinyl based monomers and crosslinkers and the subsequent hydrogel-forming chain reaction is caused by the prior formed free radicals (Fig 2). [24,25]

### Mechanical characterization

Stress-compression curves are widely used in technical studies to specify mechanical behavior like stiffness or stability for a broad range of polymers and are essential for the use in biomedical applications (e.g. as artificial tissue).[10] For comparison, all hydrogels were synthesized with a monomer concentration of 2 mol/L and a crosslinker ratio of 2 mol%. All hydrogels can be treated as a Gaussian network following Hooke's law over a specific range of deformation. In the linear phase of the compression curves, they deform resiliently and can return to their original length once the stress has been relieved. The slope of this linear segment correlates with the modulus of elasticity. Above a certain point, the sample do not return to its original

**Fig 1. Overview of the monomers used within this study** (3-sulfopropylmethacrylate potassium (MAE-SO$_3$); 3-sulfopropylacrylate potassium (AE-SO$_3$); 1-vinyl-3-ethylimidazoliumbromide (VEtImBr); (vinylbenzyl) trimethylammonium chloride (TMA-VB); [2-(acryloyloxy)ethyl]trimethylammonium chloride (AE-TMA); [2-(methacryloyloxy)ethyl]trimethylammonium chloride (MAE-TMA); [2-(methacryloyloxy)ethyl]dimethyl-(3-sulfopropyl)ammoniumhydroxide (MAEDMA-SO$_3$); 2-methacryloyloxyethylphosphorylcholine (MPC); 2-acrylamido-2-methyl-1-propanesulfonicacid (AAMPSO$_3$H);2-hydroxypropyl methacrylamide (HPMAA); hydroxyethylmethacrylate (HEMA)).

state and is referred to as break-fracture point.[10,26] Interestingly, poly(TMA-VB) could be highly compressed up to 88.8 ± 1.2% and stress values of 3.2 kPa were recorded and gained their original shape, when the stress is relieved (Fig 3). Below this fracture point poly (TMA-VB) gained their original shape under 3 min.

**Fig 2. Hydrogel synthesis via radical polymerization of a monomer and MBAA as an example for the crosslinker.**

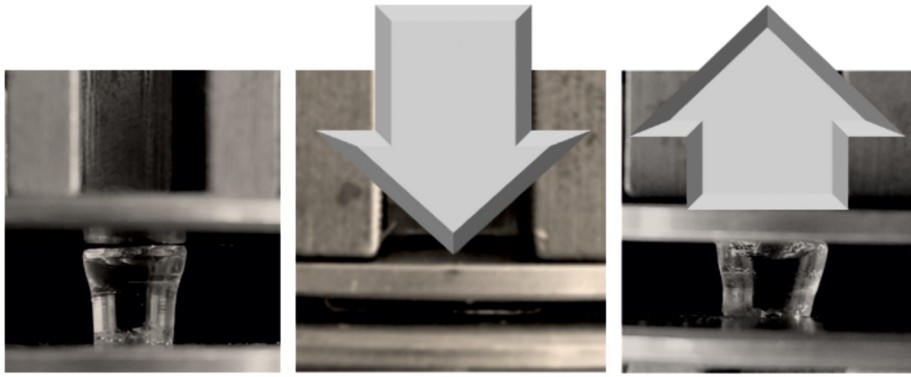

**Fig 3. Compression and release of poly (TMA-VB) [22 ± 2 ˚C; M: 2 mol/L; MBAA 2 mol%].**

The stress-compression curves for poly(VEtImBr), poly(MAE-SO$_3$), poly(MAE-TMA) and poly(AAMPSO$_3$H) hydrogels showed similar slopes (Fig 4A). Other strong deformable hydrogels were the zwitterionic poly(MPC) and poly(MAEDMA-SO$_3$) with a maximum of compression of 82.4 ± 0.8% and 83.3 ± 1.4% (Table 1, S14 and S15 Figs). In addition, the high stress resistance of poly(AE-TMA) to 5.9 kPa till break and a compression of 77.6 ± 0.9% are also taken into account. The additional methyl group of the polymerizable acryl group, being the only structural difference between poly(AE-SO$_3$) and poly(MAE-SO$_3$), strongly influences the materials properties. The hydrogel poly(MAE-SO$_3$) showed only a deformability of 59.7 ± 0.6% and a stress resistance of 1.0 kPa, whereas poly(AE-SO$_3$) showed a maximum of compression at 72.8 ± 1.7 (Fig 4A). This trend is not seen for the maximum of compression but even more pronounced for the stress-at-break for the hydrogels poly(AE-TMA) and poly(MAE-TMA). The greater steric hindrance of the hydrogels owning the additional methyl group explains this tendency. In the literature, the compression modulus of poly(HEMA) range from less than 1 kPa to greater than 500 kPa [27,28] whereas we observed a maximum of compression of 76.2 ± 2.6% and a stress resistance of 3.1 kPa (Table 1, S18 Fig). We assume that these strong differences are not only caused by the composition of the hydrogels and the structure of the different monomers but also that the shape of the specimen strongly affects the maximum of compression. The friction at the jaw inserts leads to shear stresses putting the sample into hydrostatic compression. Thus, properties at high compressive strain in compression should

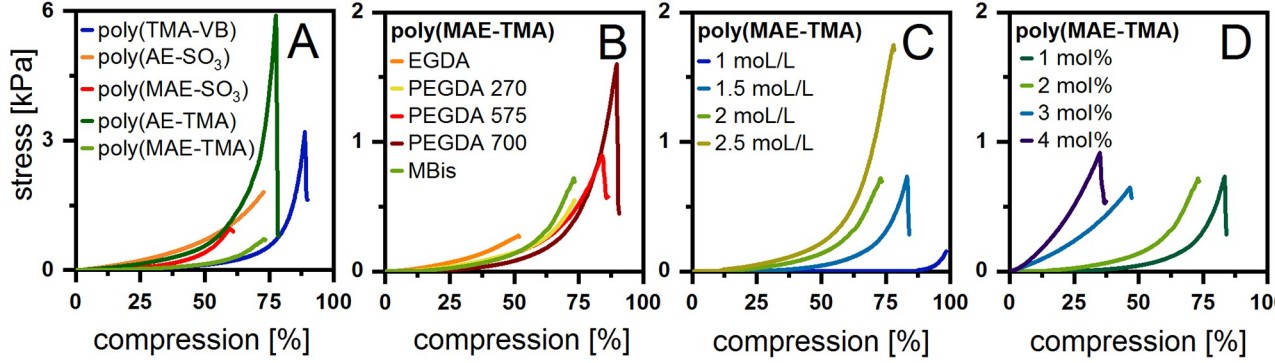

**Fig 4. Compression curves of A various hydrogels [22 ± 2 ˚C; $n \geq 3$; M: 2 mol/L; MBAA: 2 mol%]; B poly(MAE-TMA) with different CL [22 ± 2 ˚C; $n \geq 3$; M: 2 mol/L; CL: 2 mol%]; C poly(MAE-TMA) with different M amounts [22 ± 2 ˚C; $n \geq 3$; MBAA: 2 mol%] and D poly(MAE-TMA) with different MBAA amounts [22 ± 2 ˚C; $n \geq 3$; M: 2 mol/L].**

not be considered directly comparable to a true strength.[29] Therefore, the mechanical properties of hydrogels with different sample shapes are difficult to compare.

The monomer MAE-TMA was chosen for the further investigations of the hydrogel properties depending on the crosslinker and concentration of the monomer or crosslinker, since its mechanical and swelling properties are in the average of the other hydrogels in this study. Fig 4B shows the dependence of the hydrogel poly(MAE-TMA) with different crosslinker types. The mechanical properties change slightly with the lengths of the crosslinker PEGDA. The shorter the PEGDA crosslinker chain, the stiffer the obtained hydrogel. When using PEGDA 700 as a crosslinker the most compressibility (89.6 ± 0.6%) and highest force resistance (1.6 kPa) could be observed. In general, the shapes of the compression curves are very similar, indicating an influence of the type of monomer and the hydrogel composition. Fig 4C describes the influence of the monomer concentration on the mechanical properties of poly(MAE-TMA). While poly(MAE-TMA) with a monomer amount of 1 mol/L shows a maximum of compression of 98.5 ± 0.5% and a stress-at-break of 0.2 kPa, the stress-at-break increases to 1.8 kPa at 2.5 mol/L. However, the maximum of compression decreases to 78.3 ± 1.3%. In consequence, the maximum of compression is higher at a lower monomer concentration. Additionally, the mechanical properties of hydrogels can be regulated by using different crosslinker amounts. Part D Fig 4. shows the relationship between crosslinker concentration and mechanical properties. The higher the crosslinker density, the stiffer and more brittle the resulting hydrogels become. The increasing crosslinking leads to a tighter mesh tolerating less pressure.

## Degree of swelling characterization

The equilibrium swelling studies of the synthesized hydrogels from various monomer starting materials at 37°C are shown in part A of Fig 5 and were performed in distilled water. Poly (TMA-VB) reaches a degree of swelling up to 20 within 120 min. However, the equilibrium of swelling can be achieved in a range of 30 min (poly(MPC)) up to 120 min (poly(TMA-VB)) (Table 1, S20 Fig), depending on the monomer. In general, the obtained degree of swelling of the hydrogels can be divided into a strong and a slight swelling. The strongly swelling hydrogels include poly(TMA-VB) and poly(MAE-TMA) whereas poly(AE-SO$_3$), poly(MAE-SO$_3$), poly (VEtImBr), poly(AE-TMA), poly(MPC), poly(HEMA), poly(HPMAA), poly(MAEDMA-SO$_3$) and poly(AAMPSO$_3$H) are categorized as slightly swelling hydrogels. The influence of the additional methyl group is also reflected in the swelling properties of the poly(MAE-TMA) and poly(AE-TMA) as well as poly(MAE-SO$_3$) and poly(AE-SO$_3$), leading to a higher degree of swelling with an additional methyl group (Fig 5A).

Different parameters influence the hydrogel swelling, such as the thermodynamic compatibility, polymer relaxation time, the solvent motion and interaction with the functional groups of the hydrogel as well as the crosslinker chain lengths. The degree of swelling dependency of poly(MAE-TMA) on the crosslinker were compared in Fig 5B. The chemical properties of the crosslinkers have a significant effect on the swelling behavior. For instance, the hydrophilic crosslinker PEGDA is compensating the loss of hydrophilic character during the polymerization of the backbone.[30] In addition, a major factor influencing the degree of swelling has been found to be the monomer concentration in the hydrogel. Fig 5C shows the monomer dependence of poly(MAE-TMA) from 1 to 2.5 mol/L. Another dependency factor for the degree of swelling is the crosslinker amount (Fig 5D). Both the degree of swelling and the diffusion rate vary significantly. In general, the higher the crosslinker content or monomer amount, the lower the degree of swelling. In the example of poly(MAE-TMA), the degree of swelling decreases up to a crosslinker amount of 3 mol%. Interestingly, if the crosslinker concentration is further increased, the degree of swelling remains largely constant in equilibrium.

**Table 1. Comparison of the compression, the stress-at-break, the equilibrium degrees of swelling and the cell adhesion with different M and CL ratios.**

| CL | Hydrogel | M [mol%] | CL [mol%] | Compression [%] | Stress-at-break [kPa] | $q_t$ [] | Cell adhesion |
|---|---|---|---|---|---|---|---|
| MBAA | poly(AE-SO₃) | 2 | 2 | 72.8 ± 1.7 | 1.8 | 5.8 ± 0.2 | ++ |
| | poly(MAE-SO₃) | 2 | 2 | 59.7 ± 0.8 | 1.0 | 8.9 ± 0.4 | ++ |
| | poly(VEtImBr) | 2 | 2 | 53.1 ± 0.7 | 0.4 | 3.7 ± 0.1 | + |
| | poly(TMA-VB) | 2 | 2 | 88.8 ± 1.2 | 3.2 | 20.8 ± 2.8 | 0 |
| | poly(AE-TMA) | 2 | 2 | 77.6 ± 0.7 | 5.9 | 8.6 ± 0.2 | ++ |
| | poly(MPC) | 2 | 2 | 82.4 ± 0.8 | 3.2 | 3.8 ± 0.3 | + |
| | poly(MAEDMA-SO₃) | 2 | 2 | 83.3 ± 1.4 | 4.6 | 0.1 ± 0.1 | 0 |
| | poly(AAMPSO₃H) | 2 | 2 | 57.6 ± 1.8 | 1.0 | 5.9 ± 0.2 | + |
| | poly(HPMAA) | 2 | 2 | 62.6 ± 0.4 | 0.8 | 2.6 ± 0.1 | + |
| | poly(HEMA) | 2 | 2 | 76.2 ± 2.6 | 3.1 | 0.3 ± 0.2 | 0 |
| | poly(MAE-TMA) | 2 | 1 | 81.9 ± 1.3 | 1.3 | 16.8 ± 5.9 | * |
| | | 2 | 2 | 73.1 ± 0.4 | 0.7 | 13.7 ± 0.6 | + |
| | | 2 | 3 | 46.6 ± 0.6 | 0.6 | 5.3 ± 0.7 | * |
| | | 2 | 4 | 35.0 ± 0.3 | 0.9 | 4.4 ± 0.7 | * |
| | | 1 | 2 | 98.5 ± 0.5 | 0.2 | 30.2 ± 0.2 | * |
| | | 1.5 | 2 | 83.3 ± 0.7 | 0.7 | 21.0 ± 1.9 | * |
| | | 2.5 | 2 | 78.3 ± 1.3 | 1.8 | 10.9 ± 1.6 | * |
| EGDA | | 2 | 2 | 51.7 ± 2.0 | 0.3 | 13.6 ± 1.1 | * |
| PEGDA Mn = 250 | | 2 | 2 | 73.3 ± 0.6 | 0.5 | 15.1 ± 2.8 | * |
| PEGDA Mn = 575 | | 2 | 2 | 84.2 ± 0.9 | 0.9 | 21.6 ± 1.0 | * |
| PEGDA Mn = 700 | | 2 | 2 | 89.6 ± 0.6 | 1.6 | 27.6 ± 2.9 | * |

($n \geq 3$),

* = not tested, degree of swelling and compression diagrams that are not shown in the manuscript are listed in the SI.

The data for all hydrogels are compiled in Table 1 including the biocompatibility, which will be discussed in the following section.

## Preparation of the specimen and biocompatibility of the presented hydrogels

In a preliminary study we reported an excellent *in vitro* biocompatibility of a whole range of hydrogels in eluate testing approaches.[31] However, this method only addresses cytotoxic

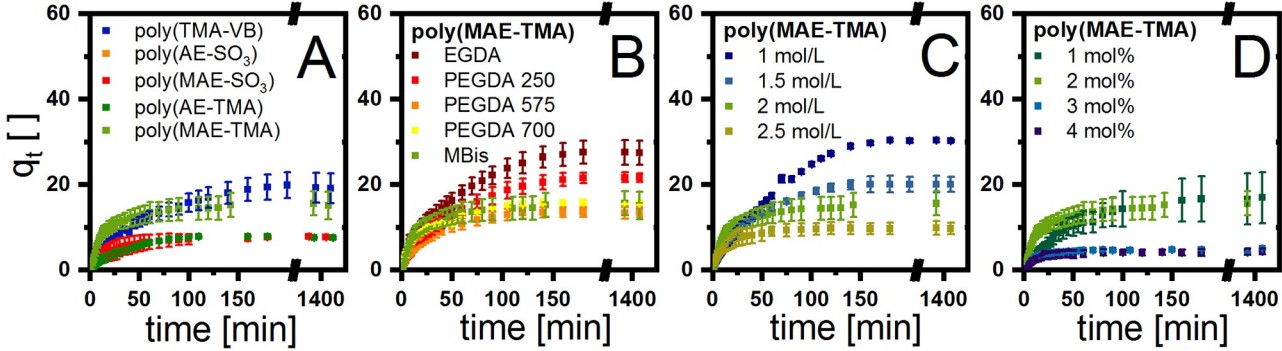

**Fig 5. Swelling kinetics of A various hydrogels [22 ± 2 ˚C; $n \geq 3$; M: 2 mol/L; MBAA: 2 mol%]; B poly(MAE-TMA) with different CL [22 ± 2 ˚C; $n \geq 3$; M: 2 mol/L; CL: 2 mol%]; C poly(MAE-TMA) with different M amounts [22 ± 2 ˚C; $n \geq 3$; MBAA: 2 mol%] and D poly(MAE-TMA) with different MBAA amounts [22 ± 2 ˚C; $n \geq 3$; M: 2 mol/L].**

effects of substances, which leach out of a material under physiological conditions and does not take the effect of the polymer surface on the cell adhesion and thus on cell growth and viability into consideration. So far, the materials to be examined have been placed on top of confluent cell layers to evaluate cell viability in direct contact.[32] In these studies, however, no conclusions can be drawn about how well a hydrogel allows for or even supports cell colonization. Such a hydrogel, utilized as a cell adherent coating on implants, might be invaluable to prevent foreign body reactions and the corresponding severe complications in various organs.[33–37] For the development of an on top direct contact test, a suitable method for the production of suitable specimens had to be devised first. The foremost requirements on the specimen were a smooth and flat surface as well as volume and shape stability. In addition, they must ensure a horizontal positioning for homogenous cell distribution. Neither the cutting of specimens from hydrogels polymerized in cylindrical form nor the direct polymerization in the 96 well plates were able to provide this. In cell cultivation experiments, hardly any cells could be cultivated on the hydrogels. The solution for this problem was the utilization of two 50 mm diameter cover slips. The hydrogel test specimens, which were polymerized between these cover glasses, had, due to the capillary forces between the glasses, the required minimum layer thickness and a sufficiently suitable smooth surface. In order to prevent volume and shape changes of the swollen specimens during the experiment, cell culture medium was used instead of distilled water as the swelling medium. In preliminary tests, the use of distilled water resulted in a change in volume and shape of the specimens in the well. The specimen shrunk so that the cells did not attach on the hydrogel but on the polystyrene beside it. The herein described developed procedure finally allowed us to evaluate cell viability quantitatively and qualitatively with an on top direct contact test technique.

In general, most of the tested hydrogels exhibit excellent biocompatibility in direct contact. With 98.4% relative cell viability poly(VEtImBr) achieved the best result in quantitative cell viability assay. The hydrogels poly(AE-SO₃), poly(MAE-SO₃), poly(MAEDMA-SO₃), poly(AAMPSO₃H), poly(HEMA) and poly(MAE-TMA) did not reduce the cell vitality by more than 20%, which corresponds to a low cytotoxicity. Concerning their cell viability these hydrogels can compete with hyaluronic acid or alginate based hydrogels exhibiting long-standing clinical use (Fig 6).[38,39] Poly(TMA-VB) poly(MPC) poly(HPMAA) exhibited with around 70% cell viability a moderate cytotoxicity.

However, the determined cell viability is an indicator for the metabolic activity of the cell on the hydrogel and therefore only one aspect for the biological evaluation. In order to carry out normal metabolism activity, proliferation and differentiation, fibroblasts must attach to and spread on an underlying matrix.[40] We addressed these questioning with qualitative microscopic analysis. On seven hydrogel surfaces we found a remarkable L929 fibroblast cell growth. Moreover the cells on poly(AE-TMA), poly(AE-SO₃) and poly(MAE-SO₃) exhibited cell morphology typically for healthy fibroblasts, indicated an outstanding cell adhesion. The hydrogels poly(TMA-VB), poly(MPC), poly(HPMAA), poly(MAE-TMA) and poly(MAEDMA-SO₃) created poor estimated cell densities and spherical shapes of the fibroblasts, indicating poor cell adhesion. The so occurring sharp differences between cell viability and the microscopic images for that hydrogels may be triggered by strong cell-cell-adhesion but weak cell-hydrogel-adhesion. This might result in detachment of the cells from the hydrogel surface due to movement of the cell culture dishes or cell culture media change. However, this only emphasizes the excellent suitability of poly(AE-TMA), poly(AE-SO₃) and poly(MAE-SO₃) as a matrix for cell growth (Fig 7).

The cell adhesion, however, is not only mediated by the biological characteristics of the cells, but also chemophysical properties of both, cells and material, play an essential role. Positively charged materials are preferred by fibroblast cells due to their negatively charged

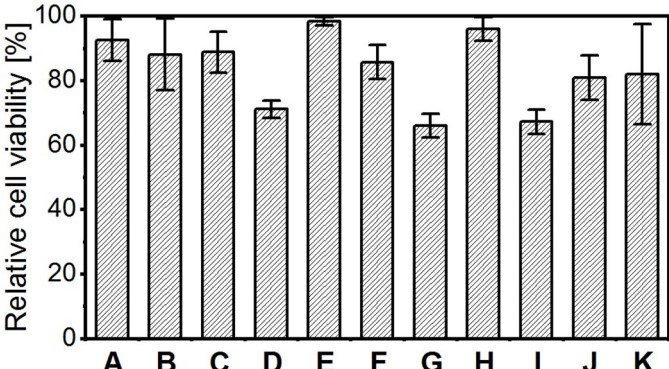

**Fig 6. Relative cell viability of A poly(AE-TMA); B poly(MAE-SO₃); C poly(AE-SO₃); D poly(MPC); E poly (VEtImBr); F poly(MAE-TMA); G poly(HPMAA); H poly(AAMPSO₃H); I poly(TMA-VB); J poly (MAEDMA-SO₃) and K poly(HEMA) was determined by the Cell Quanti Blue assay.** The relative cell vitality is determined by the ratio of the averaged fluorescence intensity in the hydrogel samples to the polystyrene negative control (100%). To support the results of the test, tetraethylthiuram disulfide (TEDT) at a concentration of $1 \cdot 10^{-4}$ mol/ L was used as the positive control, n ≥ 3.

membrane.[41] Nonetheless, several studies have demonstrated an enhanced cell adhesion and spreading on hydrophilic surfaces compared to hydrophobic surfaces.[42,43] In aqueous systems, charged materials are obviously preferable, independent from the type of charge.

Moreover, the matrix morphology plays an important role for the formation of a cell layer. Notably, the tested hydrogels with the lowest degree of swelling, such as poly(HEMA) and poly (MAEDMA-SO₃) exhibit a poor cell adhesion, In addition, we observed the best cell adhesion

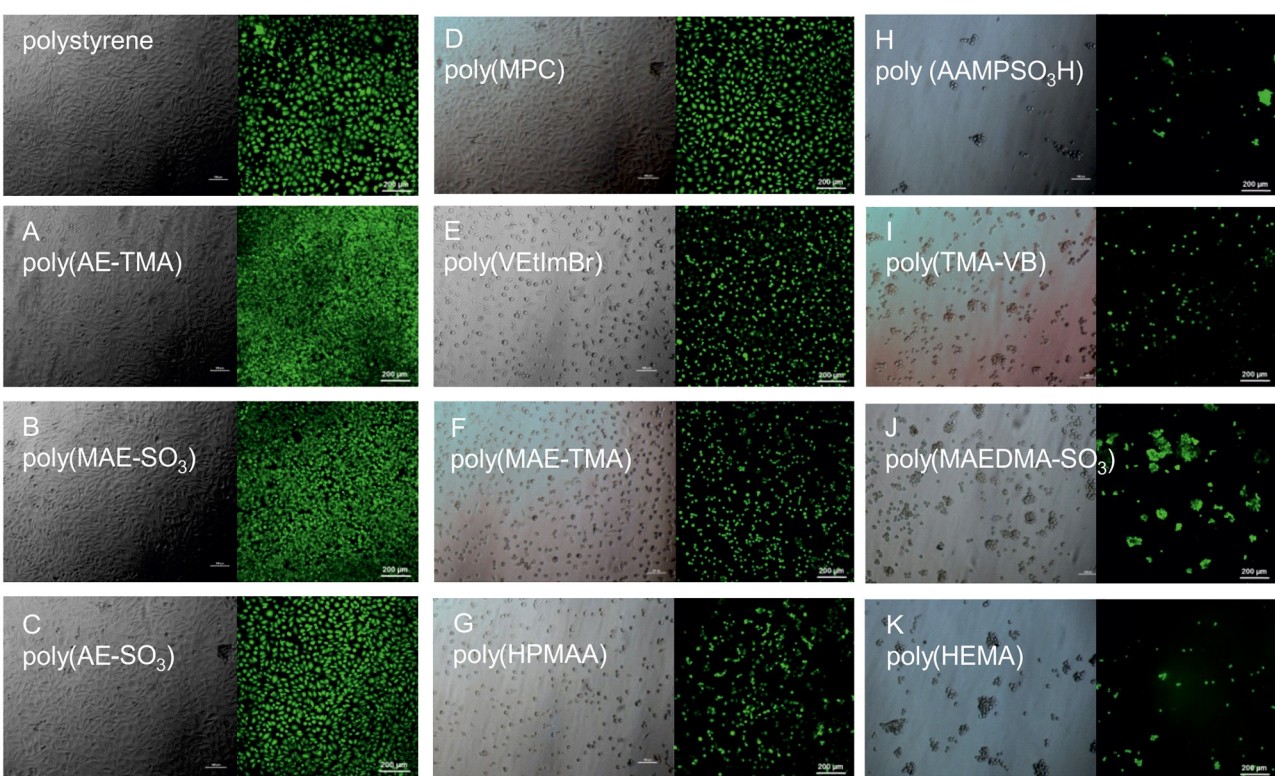

**Fig 7. Microscopic imaging with light microscopy (left) and Calcein AM staining for cytochemical analysis (right) of the hydrogels, n ≥ 3.**

on hydrogels with a swelling degree between 5 and 10 (poly(AE-SO$_3$), poly(MAE-SO$_3$), poly(AE-TMA)). In contrast to that, the hydrogels with an exceedingly high degree of swelling, such as poly(TMA-VB) have a poor cell adhesion. We assume, that the moderately swelling hydrogels form a porous surface, acting as an anchor for the fibroblast cell extensions. Moreover, a porous surface provides an enlarged surface-to-volume ratio and by this a more enhanced binding of membrane associated adhesion molecules via hydrogen bonds. Regarding strongly swelling hydrogels, the pore sizes probably exceed the appropriate size what finally results in the loss of the anchor function and subsequently leads to a reduction of hydrogen bonds.

However, when investigating materials for an application of the body, not only direct biocompatibility has to be ensured, but in addition possible toxic or harmful effects of degradation products have to be taken into account. The hydrogel materials in the presented work are based on functionalized methacrylate monomers crosslinked with MBAA. This compound class has shown no or little cytotoxic degradation products in literature.[44–46] However, since different functionalities such as sulfonic acid or amino groups may have a huge impact on the biocompatibility, especially of degradation products, this behavior has to be investigated before application testing *in vivo* and regarding potential approval processes for biomedical uses. Another aspect for potential approval processes is the immunotoxicity of the body to different kind of materials. In literature we found evidence for inflammatory processes *in vitro* and *in vivo* caused by hydrogels. Amer *et al.* reported about an toll-like receptor mediated activation of macrophages on synthetic poly (ethylene glycol) hydrogels.[47] To select suitable hydrogels for specific biomedical applications, this materials should additionally be tested for their potential to induce apoptosis.

## Summary and conclusion

Hydrogel materials derived from ionic liquids were successfully synthesized by radical polymerization of electrolytes bearing a vinyl group using MBAA and PEGDA as crosslinkers. In addition, mechanical characteristics and degree of swelling of these hydrogels and biocompatibility of the MBAA hydrogels were investigated. Our findings show that the properties of the yielded hydrogels depend on the method of preparation, polymer volume fraction, crosslinking degree, crosslinker chain length and swelling agent. The mechanical properties of the hydrogels are mainly controlled by the type of monomer and are only slightly reliant on the type of crosslinker. Desired mechanical properties are achieved by simple adjustment of the monomer and crosslinker concentration. This is practically limited by the solubility of the type of monomer and crosslinker. A new protocol to obtain hydrogel specimens for direct contact testing the biocompatibility has been shown. Direct contact testing revealed a good cell viability for the predominant part of the hydrogels competing with natural compound based polymers in longstanding clinical use.[38,39] A selection of the investigated hydrogels revealed an excellent *in vitro* biocompatibility and therefore proved their suitability for utilization in biomedical applications. Furthermore, a dependency on the degree of swelling on the cell viability has been shown. In summary, for biomedical application of hydrogels not only eluate untoxicity must been taken into account but also the effect of the surface morphology of hydrogels on the cell growth and viability.

## Supporting information

**S1 Fig. IR-spectrum of poly(AAMPSO3H) with MBAA as a crosslinker.** 3323 cm$^{-1}$ (OH, w, moisture); 2951 cm$^{-}$(C-H, d); 1650 cm$^{-1}$ (C = O, s); 1539 cm$^{-1}$ (N-H, s); 1186 cm$^{-1}$ (SO3$^{-}$, s); 1043 cm$^{-1}$ (SO3$^{-}$, s); 624 cm$^{-1}$ (C-CO-C, s) cm$^{-1}$ (C-S, w).
(TIF)

**S2 Fig. IR-spectrum of poly(AE-SO3) with MBAA as a crosslinker.** 3418 cm$^{-1}$ (OH, w, moisture); 2936 cm$^{-1}$ (C-H, d); 1720 cm$^{-1}$ (C = O, s); 1158 cm$^{-1}$ (SO3$^-$, s); 1036 cm$^{-1}$ (SO3$^-$, s); 607 cm$^{-1}$ (C-CO-C, s); 523 cm$^{-1}$ (C-S, w).
(TIF)

**S3 Fig. IR-spectrum of poly(AE-TMA) with MBAA as a crosslinker.** 3360 cm$^{-1}$ (OH, w, moisture); 2954 cm$^{-1}$ (C-H, d); 1724 cm$^{-1}$ (C = O, s); 1479 cm$^{-1}$ (CH2, s); 1158 cm$^{-1}$ (C-O, s); 952 cm$^{-1}$ (C-CO-C, s).
(TIF)

**S4 Fig. IR-spectrum of poly(HPMAA) with MBAA as a crosslinker.** 3327 cm$^{-1}$ (OH, w); 2971 cm$^{-1}$ (C-H, d);1634 cm$^{-1}$ (C = O, s); 1522 cm$^{-1}$ (N-H, s).
(TIF)

**S5 Fig. IR-spectrum of poly(MAEDMA-SO3) with MBAA as a crosslinker.** 3420 cm$^{-1}$ (OH, moisture); 2961 cm$^-$ (C-H, d); 1720 cm$^{-1}$ (C = O, s); 1648 cm$^{-1}$ (C = O, s); 1478 cm$^{-1}$ (N-H, s); 1153 cm$^{-1}$ (SO3$^-$, s); 1030 cm$^{-1}$ (SO3$^-$, s); cm$^{-1}$ (C-CO-C, s); 521 cm$^{-1}$ (C-S, w).
(TIF)

**S6 Fig. IR-spectrum of poly(MAE-SO3) with MBAA as a crosslinker.** 3441 cm$^{-1}$ (OH, w, moisture); 2944 cm$^{-1}$ (C-H, d); 1716 cm$^{-1}$ (C = O, s); 1156 cm$^{-1}$ (SO3$^-$, s); 1038 cm$^{-1}$ (SO3$^-$, s); 609 cm$^{-1}$ (C-CO-C, s); 523 cm$^{-1}$ (C-S, w).
(TIF)

**S7 Fig. IR-spectrum of poly(MAE-TMA) with MBAA as a crosslinker.** 3358 cm$^{-1}$ (OH, w, moisture); 3022 cm$^{-1}$ (C-H, d); 1718 cm$^{-1}$ (C = O, s); 1636 cm$^{-1}$ (C = O, s);1475 cm$^{-1}$ (CH2, s); 1147 cm$^{-1}$ (C-O, s); 949 cm$^{-1}$ (C-CO-C, s).
(TIF)

**S8 Fig. IR-spectrum of poly(VEtImBr) with MBAA as a crosslinker.** 3439 cm$^{-1}$ (OH, w, moisture); 2944 cm$^{-1}$ (C-H, d); 1714 cm$^{-1}$ (C = O, s); 1473 cm$^{-1}$ (N-H, s);1453 cm$^{-1}$ (CH2, s); 1156 cm$^{-1}$ (C-C-N, s); 737 cm$^{-1}$ (C-CO-C, s).
(TIF)

**S9 Fig. IR-spectrum of poly(TMA-VB) with MBAA as a crosslinker.** 3232 cm$^{-1}$ (OH, w, moisture); 2963 cm$^{-1}$ (= C-H, m); 1642 cm$^{-1}$ (-C = C, s); 1477 cm$^{-1}$ (-C = C, s); 1261 cm$^{-1}$ (CH3, s); 1092 cm$^{-1}$ (C -N, s); 890 cm$^{-1}$ (= C-H, s).
(TIF)

**S10 Fig. IR-spectrum of poly(MPC) with MBAA as a crosslinker.** 3340 cm$^{-1}$ (OH, w, moisture); 1716 cm$^{-1}$ (C-O, d); 1479 cm$^{-1}$ (CH3, s); 1228 cm$^{-1}$ (C-C-N, s); 1057 cm$^{-1}$ (P-O, s); 954 cm$^{-1}$ (C-N, s); 921 cm$^{-1}$ (P-O, s); 783 cm$^-$ (C-CO-C, s).
(TIF)

**S11 Fig. IR-spectrum of poly(HEMA) with MBAA as a crosslinker.** 3336 cm$^{-1}$ (OH, w); 2961 cm$^{-1}$ (C-H, d); 1706 cm$^{-1}$ (C = O, s); 1632 cm$^{-1}$ (C = O, s); 1448 cm$^{-1}$ (CH2, s); 1261 cm$^{-1}$ (C-O-C, s); 1158 cm$^{-1}$ (CH3, s); 1073 cm$^{-1}$ (); 892 cm$^{-1}$ (C-CO-C, s).
(TIF)

**S12 Fig. IR-spectrum of poly(MAE-TMA) with 700 PEGDA as a crosslinker.** 3362 cm$^{-1}$ (OH, w, moisture);2963 cm$^{-1}$ (C-H, d); 1720 cm$^{-1}$ (C = O, s); 1634 cm$^{-1}$ (C = O, s);1475 cm$^{-1}$ (CH2, s); 1230 cm$^{-1}$ (C-O, s); 947 cm$^{-1}$ (C-CO-C, s).
(TIF)

**S13 Fig. Compression curve of poly(VEtImBr) with MBAA as a crosslinker** [22 ± 2 ˚C; n ≥ 3; M: 2 mol/L; MBAA: 2 mol%].
(TIF)

**S14 Fig. Compression curve of poly(MAEDMA-SO3) with MBAA as a crosslinker** [22 ± 2 ˚C; n ≥ 3; M: 2 mol/L; MBAA: 2 mol%].
(TIF)

**S15 Fig. Compression curve of poly(MPC) with MBAA as a crosslinker** [22 ± 2 ˚C; n ≥ 3; M: 2 mol/L; MBAA: 2 mol%].
(TIF)

**S16 Fig. Compression curve of poly(AAMPSO3H) with MBAA as a crosslinker** [22 ± 2 ˚C; n ≥ 3; M: 2 mol/L; MBAA: 2 mol%].
(TIF)

**S17 Fig. Compression curve of poly(HPMAA) with MBAA as a crosslinker** [22 ± 2 ˚C; n ≥ 3; M: 2 mol/L; MBAA: 2 mol%].
(TIF)

**S18 Fig. Compression curve of poly(HEMA) with MBAA as a crosslinker** [22 ± 2 ˚C; n ≥ 3; M: 2 mol/L; MBAA: 2 mol%].
(TIF)

**S19 Fig. Compression curve of poly(VEtImBr) with MBAA as a crosslinker** [22 ± 2 ˚C; n ≥ 3; M: 2 mol/L; MBAA: 2 mol%].
(TIF)

**S20 Fig. Compression curve of poly(MPC) with MBAA as a crosslinker** [22 ± 2 ˚C; n ≥ 3; M: 2 mol/L; MBAA: 2 mol%].
(TIF)

**S21 Fig. Compression curve of poly(AAMPSO3H) with MBAA as a crosslinker** [22 ± 2 ˚C; n ≥ 3; M: 2 mol/L; MBAA: 2 mol%].
(TIF)

**S22 Fig. Compression curve of poly(HPMAA) with MBAA as a crosslinker** [22 ± 2 ˚C; n ≥ 3; M: 2 mol/L; MBAA: 2 mol%].
(TIF)

**S23 Fig.**
(TIF)

**S1 Data.**
(PDF)

## Acknowledgments

We thank Katrin Feest, Jan von Langermann and Lars-Erik Meyer for proof reading this paper. We also thank Sandra Diederich and Daniela Arbeiter for their technical support.

## Author Contributions

**Conceptualization:** Udo Kragl.

**Data curation:** Johanna Claus, Andreas Brietzke.

**Formal analysis:** Celina Lehnert.

**Methodology:** Andreas Brietzke.

**Supervision:** Niels Grabow, Udo Kragl.

**Writing – original draft:** Johanna Claus, Andreas Brietzke.

**Writing – review & editing:** Stefan Oschatz, Niels Grabow.

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
