## [Decision Letter · Decision Letter 0]

7 Feb 2020

PONE-D-19-34741

Swelling characteristics and biocompatibility of ionic liquid based hydrogels for biomedical applications

PLOS ONE

Dear Prof. Dr. Kragl,

Thank you for submitting your manuscript to PLOS ONE. After careful consideration, we feel that it has merit but does not fully meet PLOS ONE’s publication criteria as it currently stands. Therefore, we invite you to submit a revised version of the manuscript that addresses the points raised during the review process.

We would appreciate receiving your revised manuscript by Mar 23 2020 11:59PM. To enhance the reproducibility of your results, we recommend that if applicable you deposit your laboratory protocols in protocols.io, where a protocol can be assigned its own identifier (DOI) such that it can be cited independently in the future. For instructions see: http://journals.plos.org/plosone/s/submission-guidelines#loc-laboratory-protocols

We look forward to receiving your revised manuscript.

Kind regards,

Feng Zhao

Academic Editor

PLOS ONE

**Comments to the Author**

1. Is the manuscript technically sound, and do the data support the conclusions?

Reviewer #1: Yes

Reviewer #2: Partly

2. Has the statistical analysis been performed appropriately and rigorously? 

Reviewer #1: Yes

Reviewer #2: No

3. Have the authors made all data underlying the findings in their manuscript fully available?

Reviewer #1: Yes

Reviewer #2: Yes

4. Is the manuscript presented in an intelligible fashion and written in standard English?

Reviewer #1: Yes

Reviewer #2: Yes

5. Review Comments to the Author

Reviewer #1: This research paper prepared various types of polyionic hydrogels using the same crosslinkers and initiators. The difference of mechanical properties, swelling ratio and biocompatibility among different composition of hydrogels were compared. The correlation of mechanical properties and biocompatibility to composition, cross-linkers and monomers concentrations of hydrogels were assessed. This paper provided a good mechanism reference for hydrogels synthesis and their potential applications.

Comments for minor revisions include spell checking (typos happened, e.g. line 239 should be "cells did not "attach" on the hydrogel"); supplementary results were not described in the main text.

Reviewer #2: In the current work, hydrogel materials derived from ionic liquids were successfully synthesized by radical polymerization of electrolytes bearing a vinyl group using MBAA and PEGDA as crosslinkers. The mechanical characteristics, degree of swelling and biocompatibility of the MBAA hydrogels were investigated. Although impressive, following issues needs to be addressed to make the manuscript technically sound.

1. The information regarding statistical analysis is missing.

2. Line 122: Please include the seeding density of the L929 mouse fibroblast per well of the 96-well plate.

3. Line 112: How much hydrogel (or how many milliliters) was added between two 50 mm diameter cover glass for the biocompatibility assay?

4. Clearly explain how the dead cells were quantified? and compared with which CONTROL to calculate relative cell viability (Fig. 6).

5. Line 256-257: "The hydrogels poly(TMA-VB), poly(MPC), poly(HPMAA), poly(MAE-TMA) and poly(MAEDMA-SO3) created poor cell counts and spherical shapes of the fibroblasts, indicating poor cell adhesion" : Explain how the cells were counted and compared with which control?

Here, the poor cell counts can be result from two reasons: (1) Lower cellular adhesion property of the hydrogel itself (due to difference in porosity or swelling, as discussed by the authors) and/or (2) Hydrogel induced cytotoxicity that allows cells to initially attach on the hydrogel surface, BUT due to the cytotoxic hydrogel components, cells undergo apoptosis and finally detach from the hydrogel surface. To prove that the hydrogel ITSELF is not cytotoxic, it is important to perform ethidium homodimer-1 (EtHD-1) staining along with calcein-AM. EtHD-1 can detect the hydrogel-attached apoptotic cells to determine whether hydrogel itself is cytotoxic or not. Please address and discuss this issue.

6. No information regarding degradation rate of these hydrogel is provided. Since these hydrogels are proposed to be used for biomedical applications, it is very critical to study their rate of degradation.

7. Please include discussion regarding the immune cell response (especially macrophages) toward these hydrogels if previously reported in the literature. Importantly, discuss whether the degradation products of these hydrogel can induce severe immune response.

6. PLOS authors have the option to publish the peer review history of their article (what does this mean?). If published, this will include your full peer review and any attached files.

Reviewer #1: No

Reviewer #2: No

---

## [Author Response · Author response to Decision Letter 0]

24 Feb 2020

Response to Reviewer

Swelling characteristics and biocompatibility of ionic liquid based hydrogels for biomedical applications

PONE-D-19-34741

Reviewer #1

Reviewer #1: This research paper prepared various types of polyionic hydrogels using the same crosslinkers and initiators. The difference of mechanical properties, swelling ratio and biocompatibility among different composition of hydrogels were compared. The correlation of mechanical properties and biocompatibility to composition, cross-linkers and monomers concentrations of hydrogels were assessed. This paper provided a good mechanism reference for hydrogels synthesis and their potential applications.

Comments for minor revisions include spell checking (typos happened, e.g. line 239 should be "cells did not "attach" on the hydrogel"); supplementary results were not described in the main text.

Response from the authors: Thank you for this useful hint. Some spelling mistakes where found and corrected. Additional references to the Supporting information have been added, e.g. line 164, line 171, line 195-196, line 200-201.

Reviewer #2

Reviewer #2: In the current work, hydrogel materials derived from ionic liquids were successfully synthesized by radical polymerization of electrolytes bearing a vinyl group using MBAA and PEGDA as crosslinkers. The mechanical characteristics, degree of swelling and biocompatibility of the MBAA hydrogels were investigated. Although impressive, following issues needs to be addressed to make the manuscript technically sound.

1. The information regarding statistical analysis is missing.

Response from the authors: Thank you for this comment.

We have modified the manuscript accordingly. In line 127-128 we added the sentences: “Replicates (n ≥ 3) were tested for normal distribution and subjected to a Nalimovs’ test for outliers. Means and standard errors (SEM) were calculated in Sigma Plot.” In addition, the missing number of replicates (n ≥ 3) are added to Fig. 6 and Fig. 7.

2. Line 122: Please include the seeding density of the L929 mouse fibroblast per well of the 96-well plate.

Response from the authors: Thank you for this comment. We added this information in our manuscript at line 122-123: “1x104 L929 mouse fibroblasts were seeded directly on the hydrogel specimen with 200 µL culture medium per well and incubated under cell culture conditions (37°C, 5% CO2) for 46 hours (fibroblast cell density 62500 cells x cm2).”

3. Line 112: How much hydrogel (or how many milliliters) was added between two 50 mm diameter cover glass for the biocompatibility assay?

Response from the authors: The information has been added to the materials and methods part of the manuscript (line 112): “For biocompatibility testing, 1 mL of the monomer-crosslinker solution was polymerized between two 50 mm diameter cover slips to obtain specimen with the maximum possible surface.”

4. Clearly explain how the dead cells were quantified? and compared with which CONTROL to calculate relative cell viability (Fig. 6).

Response from the authors: Thank you for this hint. We extended the legend of Figure 6 in line 251 to: “Relative cell viability of A poly(AE-TMA); B poly(MAE-SO3); C poly(AE-SO3); D poly(MPC); E poly(VEtImBr); F poly(MAE-TMA); G poly(HPMAA); H poly(AAMPSO3H); I poly(TMA-VB); J poly(MAEDMA-SO3) and K poly(HEMA) was determined by the Cell Quanti Blue assay. The relative cell viability is determined by the ratio of the averaged fluorescence intensity in the hydrogel samples to the polystyrene negative control (100%). To support the results of the test, tetraethylthiuram disulfide (TETD) at a concentration of 1х10-4 mol/L was used as the positive control.” Another part of the description has been removed as it refers to the synthesis conditions of the hydrogels.

Additionally we enhanced line 131: “Cells were stained with Calcein AM (Thermo Fisher Waltham, MA, USA) to prove the viability of the living cells.”

Additionally, we changed line 263-264 to: “The hydrogels poly(TMA-VB), poly(MPC), poly(HPMAA), poly(MAE-TMA) and poly(MAEDMA-SO3) created poor estimated cell densities and spherical shapes of the fibroblasts, indicating poor cell adhesion.”

5. Line 256-257: "The hydrogels poly(TMA-VB), poly(MPC), poly(HPMAA), poly(MAE-TMA) and poly(MAEDMA-SO3) created poor cell counts and spherical shapes of the fibroblasts, indicating poor cell adhesion" : Explain how the cells were counted and compared with which control?

Here, the poor cell counts can be result from two reasons: (1) Lower cellular adhesion property of the hydrogel itself (due to difference in porosity or swelling, as discussed by the authors) and/or (2) Hydrogel induced cytotoxicity that allows cells to initially attach on the hydrogel surface, BUT due to the cytotoxic hydrogel components, cells undergo apoptosis and finally detach from the hydrogel surface. To prove that the hydrogel ITSELF is not cytotoxic, it is important to perform ethidium homodimer-1 (EtHD-1) staining along with calcein-AM. EtHD-1 can detect the hydrogel-attached apoptotic cells to determine whether hydrogel itself is cytotoxic or not. Please address and discuss this issue.

Response from the authors: 

Thank you very much for your detailed comment. Our microscopic analysis did not aim to provide a quantitative conclusion on cytotoxicity. Rather, it was intended to support the quantitative analysis with the CellQuanti Blue Assay. The microscopic images should prove that the cells were actually actively spreading on the hydrogel, displaying another quality of the hydrogels besides cell viability. In earlier experiments using a different preparation methods of the specimens, we found that the cells slipped off the hydrogel and grew on the cell culture polystyrene. Very good cell viability could be measured here as well. 

Furthermore, we were able to make a qualitative evaluation of the cell morphology by microscopic analysis. The Calcein AM staining was also intended to demonstrate that the cells on the hydrogels are viable, but not to provide a quantification for the evaluation of cytotoxicity. The question of whether apoptotic or possibly also necrotic processes result in a reduction of cell count and cell viability is undoubtedly very interesting, but not the subject of this study. Rather, we intended to pre-select candidates from a whole range of hydrogels based on basic parameters. After selecting generally suitable hydrogels for specific biomedical applications, this materials should indeed be tested for their potential to induce apoptosis.

We added another paragraph to our manuscript (line 293-294): “To select suitable hydrogels for specific biomedical applications, this materials should additionally be tested for their potential to induce apoptosis.”

6. No information regarding degradation rate of these hydrogel is provided. Since these hydrogels are proposed to be used for biomedical applications, it is very critical to study their rate of degradation.

Response from the authors: The degradation of these hydrogels is certainly interesting and necessary for this type of application. We added a paragraph to our manuscript regarding this topic (line 285-290): “However, when investigating materials for an application of the body, not only direct biocompatibility has to be ensured, but in addition possible toxic or harmful effects of degradation products have to be taken into account. The hydrogel materials in the presented work are based on functionalized methacrylate monomers crosslinked with MBAA. This compound class has shown no or little cytotoxic degradation products in literature (44–46). However, since different functionalities such as sulfonic acid or amino groups may have a huge impact on the biocompatibility, especially of degradation products, this behavior has to be investigated before application testing in vivo and regarding potential approval processes for biomedical uses.”

However, these experiments and results would go beyond the scope of this manuscript but will be addressed in the ongoing work on this topic. 

7. Please include discussion regarding the immune cell response (especially macrophages) toward these hydrogels if previously reported in the literature. Importantly, discuss whether the degradation products of these hydrogel can induce severe immune response.

Response from the authors: 

Thank you for this interesting hint. In our response to comment 5 we already claimed that we intended to pre-select hydrogels for specific biomedical applications based on basic parameters. Nevertheless, aspects of the immunotoxicity are very interesting and must be addressed within approval process for biomedical applications. Indeed, we found evidence for inflammatory processes in vitro and in vivo caused by hydrogels in literature. For example Amer et al., 2019 found an Toll-like receptor mediated activation of macrophages on synthetic poly (ethylene glycol) hydrogels. We appreciate your advice and will consider this in our future studies.

We added a paragraph to our manuscript regarding this topic (line 290 – 293): “Another aspect for potential approval processes is the immunotoxicity of the body to different kind of materials. In literature we found evidence for inflammatory processes in vitro and in vivo caused by hydrogels. Amer et al. reported about an toll-like receptor mediated activation of macrophages on synthetic poly (ethylene glycol) hydrogels (47).”

---

## [Decision Letter · Decision Letter 1]

24 Mar 2020

Swelling characteristics and biocompatibility of ionic liquid based hydrogels for biomedical applications

PONE-D-19-34741R1

Dear Dr. Kragl,

We are pleased to inform you that your manuscript has been judged scientifically suitable for publication and will be formally accepted for publication once it complies with all outstanding technical requirements.

With kind regards,

Feng Zhao

Academic Editor

PLOS ONE

**Comments to the Author**

1. If the authors have adequately addressed your comments raised in a previous round of review and you feel that this manuscript is now acceptable for publication, you may indicate that here to bypass the “Comments to the Author” section, enter your conflict of interest statement in the “Confidential to Editor” section, and submit your "Accept" recommendation.

Reviewer #1: All comments have been addressed

Reviewer #2: All comments have been addressed

2. Is the manuscript technically sound, and do the data support the conclusions?

Reviewer #1: Yes

Reviewer #2: Yes

3. Has the statistical analysis been performed appropriately and rigorously? 

Reviewer #1: Yes

Reviewer #2: I Don't Know

4. Have the authors made all data underlying the findings in their manuscript fully available?

Reviewer #1: Yes

Reviewer #2: Yes

5. Is the manuscript presented in an intelligible fashion and written in standard English?

Reviewer #1: Yes

Reviewer #2: Yes

6. Review Comments to the Author

Reviewer #1: (No Response)

Reviewer #2: (No Response)

---

## [Editor Report · Acceptance letter]

7 Apr 2020

PONE-D-19-34741R1 

Swelling characteristics and biocompatibility of ionic liquid based hydrogels for biomedical applications 

Dear Dr. Kragl:

I am pleased to inform you that your manuscript has been deemed suitable for publication in PLOS ONE. Congratulations! Your manuscript is now with our production department. 

With kind regards,

on behalf of

Dr. Feng Zhao 

Academic Editor

PLOS ONE